# Quantitative Checklist for Autism in Toddlers (Q-CHAT). A population screening study with follow-up: the case for multiple time-point screening for autism

Carrie Allison [1], Fiona E Matthews [2], Liliana Ruta,[3] Greg Pasco,[4] Renee Soufer,[1] Carol Brayne,[5] Tony Charman,[4] Simon Baron-Cohen[1]

For numbered affiliations see end of article.

**Correspondence to**
Dr Carrie Allison; cla29@cam.ac.uk

## ABSTRACT

**Objective** This is a prospective population screening study for autism in toddlers aged 18–30 months old using the Quantitative Checklist for Autism in Toddlers (Q-CHAT), with follow-up at age 4.

**Design** Observational study.

**Setting** Luton, Bedfordshire and Cambridgeshire in the UK.

**Participants** 13 070 toddlers registered on the Child Health Surveillance Database between March 2008 and April 2009, with follow-up at age 4; 3770 (29%) were screened for autism at 18–30 months using the Q-CHAT and the Childhood Autism Spectrum Test (CAST) at follow-up at age 4.

**Interventions** A stratified sample across the Q-CHAT score distribution was invited for diagnostic assessment (phase 1). The 4-year follow-up included the CAST and the Checklist for Referral (CFR). All with CAST ≥15, phase 1 diagnostic assessment or with developmental concerns on the CFR were invited for diagnostic assessment (phase 2). Standardised diagnostic assessment at both time-points was conducted to establish the test accuracy of the Q-CHAT.

**Main outcome measures** Consensus diagnostic outcome at phase 1 and phase 2.

**Results** At phase 1, 3770 Q-CHATs were returned (29% response) and 121 undertook diagnostic assessment, of whom 11 met the criteria for autism. All 11 screened positive on the Q-CHAT. The positive predictive value (PPV) at a cut-point of 39 was 17% (95% CI 8% to 31%). At phase 2, 2005 of 3472 CASTs and CFRs were returned (58% response). 159 underwent diagnostic assessment, including 82 assessed in phase 1. All children meeting the criteria for autism identified via the Q-CHAT at phase 1 also met the criteria at phase 2. The PPV was 28% (95% CI 15% to 46%) after phase 1 and phase 2.

**Conclusions** The Q-CHAT can be used at 18–30 months to identify autism and enable accelerated referral for diagnostic assessment. The low PPV suggests that for every true positive there would, however, be ~4–5 false positives. At follow-up, new cases were identified, illustrating the need for continued surveillance and rescreening at multiple time-points using developmentally sensitive instruments. Not all children who later receive a diagnosis of autism are detectable during the toddler period.

### What is known about the subject?

► In the UK, a systematic population screening programme for autism is not recommended to facilitate early detection. This is because general population screening tests that have been evaluated using systematic follow-up have not proved accurate.

### What this study adds?

► It is possible to detect autism at 18–30 months.
► It is not possible to identify every child at a very young age who will later be diagnosed with autism.
► The Quantitative Checklist for Autism in Toddlers can be used to screen toddlers at 18–30 months.

## INTRODUCTION

Autism affects approximately 1%–2% of the population.[1 2] Screening requires a standardised and systematic approach to identify children who may require a diagnostic assessment. The costs and benefits of autism population screening need to be carefully balanced. Undue anxiety in false positives and the need for sufficient support services for those diagnosed should be offset by the many benefits of accurate screening that include earlier diagnosis, more timely access to specific interventions[3] and better support for parents. In the UK, no autism-specific screening instrument is recommended by the National Institute for Health and Care Excellence.[4] Autism is variable in onset and significance over time of specific symptoms. No existing screening instrument, used at a single time-point, reaches a minimum of 80% sensitivity and specificity.[5] In contrast, the American Academy of Pediatrics[6] recommends routine screening at well-child checks using

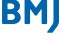

the Checklist for Autism in Toddlers (CHAT)[7 8] and the Modified Checklist for Autism in Toddlers (M-CHAT),[9] despite the limited research evidence for their use[10] and insufficient evidence to determine if certain risk factors modify the performance characteristics of the tests.[11]

Lengthy delays between the first concerns and an eventual diagnosis are often experienced.[12] There has been no evidence of a reduction in the age at diagnosis over the past decade in the UK.[13] Beliefs held by health professionals about screening for autism (eg, system capacity, interventions available) may influence practice.[14] Equally, young children with autism may not present salient atypical behaviour in a short consultation to warrant an assessment referral. Atypical behaviours, such as restricted and repetitive behaviours, extend to the neurotypical child on a continuum[15] and may reduce with time.[16] The reduction of inappropriate referrals to already overstretched services could serve to benefit children, their families and health services. Of the referrals to a child development centre in the UK for possible autism, 60% resulted in an autism diagnosis. It is unclear what the outcome was for the remainder of the referrals.[17]

The CHAT[7] is a short parent report questionnaire combined with a health professional observation. The CHAT was tested prospectively in 18-month-old toddlers from the general population, with a diagnostic follow-up of those who scored above the cut-point to allow estimation of both sensitivity and specificity. At 18 months the CHAT identified 92% of children with autism when assessed and diagnosed by two independent psychiatrists using the Diagnostic and Statistical Manual of Mental Disorders, Third Edition, Revised criteria as the gold standard at that time.[8] By 6-year follow-up, the sensitivity was 38%, indicating that many children did not present with sufficient features of autism to meet the diagnostic criteria at 18 months. Nevertheless, the CHAT had very high specificity (98%) and was the first demonstration internationally that autism could be diagnosed as early as 18 months of age, when previously age at first diagnosis was typically not until later in childhood.[18] The CHAT was based on toddler developmental precursors of 'theory of mind',[19] such as joint attention[20–22] and pretend play,[23] and so stemmed from a cognitive developmental theory about the nature of autism.[24]

The M-CHAT[9] is a 23-item parent report questionnaire used in the US healthcare system. It incorporates the original nine items from the CHAT and includes additional items that may be characteristic of a young child with autism. Initial results reported a sensitivity of 0.97, a specificity of 0.99 and a positive predictive value (PPV) of 0.68 when used with a telephone follow-up. Despite limitations to the initial studies[25] (eg, referred and unselected samples were combined and no systematic follow-up), the M-CHAT has been examined in a large population sample at 18 months.[26] Specificity was high (0.93), but sensitivity was low (0.34) and PPV was 0.35. The authors concluded that it may not be possible to identify all children with autism at 18 months. The M-CHAT has subsequently

been revised (M-CHAT, Revised with Follow-Up),[27] consisting of 20 items and including examples to provide developmental context and clarity. This was tested in a large population (n=16071), finding PPV to be 0.48, while another study[28] reported PPV to be 0.54. Sensitivity could not be determined in these two studies because only the screen positives were followed up. An additional study examined the accuracy of the Modified Checklist for Autism in Toddlers with Follow-Up (M-CHAT/F) in a universal, primary care-based screening context that involved systematic follow-up up to 8 years.[29] Diagnoses were ascertained independent of the screen. The study reported sensitivity of 38.8% and PPV of 14.6%, with almost universal screening being achieved (91%) by incorporating autism screening into routine check-ups and integrating these within the electronic health record.

The Quantitative Checklist for Autism in Toddlers (Q-CHAT)[30] was developed to dimensionalise autism in toddlerhood. It has been demonstrated that autistic traits can be measured quantitatively in the general population, using the Autism Spectrum Quotient (AQ) in adulthood,[31] the AQ-Adolescent,[32] the AQ-Child[33] and the Social Responsiveness Scale,[34] in older children and adults. The Q-CHAT is a parent report 25-item questionnaire that includes the CHAT and additional items. Each item is converted to a rating scale, thus quantifying autistic traits. This allows for the possibility of reporting behaviour at a reduced frequency. For example, the response options on the item concerning protodeclarative pointing range from many times a day (least autistic response) through to never (most autistic response). The Q-CHAT domains include social communication, repetitive, stereotyped and sensory behaviours. A preliminary study examined the distribution of the Q-CHAT in an unselected sample of toddlers aged 18–24 months old and in a sample of children already diagnosed on the autism spectrum. Results indicated that the Q-CHAT discriminated well between young children with autism and unselected toddlers in the population at 18–24 months.[30] The Q-CHAT was not evaluated as a prospective screen in this sample. The Q-CHAT has been translated and validated in other countries, including Italy and Chile.[35 36] A short version of the Q-CHAT (Q-CHAT-10)[37] was developed and tested retrospectively on toddlers with an autism diagnosis compared with toddlers from the unselected population reported. The sensitivity and specificity were 0.91 and 0.89, respectively. However, these studies were not longitudinal and so they cannot address the question of whether it is best to screen for autism at a single or multiple time-points. See Petrocchi et al[38] for a recent systematic review outlining the psychometric properties of autism-specific screening instruments.

The objective of this study is to report a population screening study of nearly 4000 toddlers at 18–30 months using the Q-CHAT, with diagnostic assessments on a subsample of responders and subsequent follow-up at 4 years. The Child Health Surveillance Database was used to identify the population eligible to screen with the

Q-CHAT. The study is reported in two phases, as undertaken. The aims of phase 1 were (1) to determine the test accuracy of the Q-CHAT in the toddler period and (2) to determine an optimal screening cut-point that could be used to select toddlers for a diagnostic assessment. The aims of phase 2 were (1) to rescreen the population at a minimum age of 4 using the Childhood Autism Spectrum Test (CAST)[39] in order to identify children with autism who were not identified at phase 1 (false negatives) and the Checklist for Referral (CFR) to seek information on the child's history of developmental concerns; (2) to confirm the outcome of those who were identified by the Q-CHAT at 18–30 months, and those who were not, by conducting diagnostic assessments; (3) to assess the stability of the diagnostic outcome from phase 1 to phase 2; and (4) to assess the discriminant power of the Q-CHAT and CAST and CFR in distinguishing autism cases from non-autism cases. This two-phase design allowed us to report sensitivity, specificity and PPV. Test accuracy at phase 1 was not evaluated at the time to ensure researchers remained blind to the results when undertaking phase 2.

## METHODS
### Patient and public involvement
The first and senior authors (CA and SB-C) gave talks to parent support groups and clinicians about this research during all phases of the research and received feedback through questions and discussion. The format of the Q-CHAT itself was co-designed with a parent support group.

### Setting, study design and procedure
#### Phase 1
All caregivers of infants between the ages of 18 and 30 months who were registered on the Child Health Surveillance Database at the three primary care trusts (PCT) on the date of mailing were invited to complete the Q-CHAT between March 2008 and April 2009. Questionnaires were sent in three batches to manage capacity of the team. Screening was conducted via a postal questionnaire, distributed by the PCT. Luton questionnaires were mailed in March 2008, Bedfordshire questionnaires were mailed in May 2008, and Cambridgeshire questionnaires were mailed in April 2009. Questionnaires were posted twice, 2 weeks apart, to maximise response. Returned questionnaires were scored by the research team and the sampling strategy, as detailed in the next section, was applied to establish the diagnostic assessment sample. Research diagnostic assessments were arranged as soon as possible after completion of the Q-CHAT. All members of the diagnostic assessment team remained blind to the child's Q-CHAT score. The Q-CHAT was completed a second time prior to any of the diagnostic assessment battery in order to later examine the test–retest reliability of the Q-CHAT in a sample enriched with high scorers, which will be reported separately. Diagnostic outcome

was defined by the researchers reaching a consensus diagnosis using all diagnostic assessment data and researcher judgement.

### Phase 1 sampling strategy
The strategy for selecting participants for follow-up diagnostic assessment was weighted towards those with higher scores. The rationale for choosing 44 as the cut-point for inviting participants for follow-up diagnostic assessment at phase 1 was based on anticipated estimates of the prevalence of autism (approximately 1%) balanced with the knowledge that over 70% of children who already had a diagnosis on the autism spectrum scored 44 and above in our initial study.[30] A cut-point of 44 was therefore chosen in order to maximise sensitivity while generating a feasible number of diagnostic assessments to be completed. In contrast, the strategy for identifying the optimal screening cut-point on the Q-CHAT was not determined until after phase 2 diagnostic assessments were complete, which was one of the aims of the study. At a cut-point of 44 on the Q-CHAT at phase 1, 100% of children were selected for diagnostic assessment. To ensure that children with missing data were not missed from being included in a high scoring group (44+), missing Q-CHAT items were allocated the maximum item score (4) and the total Q-CHAT score recalculated. The rationale for incorporating missing data into the sampling strategy was to ensure that children with a high likelihood of having many autistic traits (and therefore potentially being autistic) but who had sufficient missing data to put them in a sampling band where the chance of being selected was low were included in the follow-up diagnostic assessments. Each Q-CHAT questionnaire therefore had two scores: the observed (assuming each item of missing data would score as not impaired) and the maximum (assuming each item of missing data would score as impaired). Observed and maximum Q-CHAT scores were grouped into four bands (≥44, 41–43, 38–40, ≤37). This ensured that missing data meant a child was more likely to be assessed.

The final sampling groups were determined according to the maximum score, taking into account the movement across score groups from observed score to maximum score. For example, if a child's observed score was 40 and there were two missing Q-CHAT items, 8 was added to the observed score to give a score of 48, which would therefore be that child's sampling score. Sampling group allocation and randomisation took place prior to establishing whether or not the family had consented for further contact.

When the chance of being selected was less than 100%, selection for diagnostic assessment was undertaken using a random number generator (www.random.org). One per cent of the children who had a maximum score ≤37 were selected for diagnostic assessment to ensure that the diagnostic assessment team were blind to whether there were likely to be concerns about the child at the time of diagnostic assessment.

## Phase 2

All caregivers who consented to further contact following the phase 1 screening (n=3472) were invited to participate in phase 2 at a minimum age of 48 months, between January and July 2011. Caregivers were sent the CAST, a 37-item tool to detect autism in children among those aged 4–11 years, in a non-clinical setting.[39 40] Parents were also sent a brief checklist enquiring about whether the child had ever been referred or diagnosed with any developmental and/or medical condition (CFR).

### Phase 2 sampling

All children who scored ≥15 on the CAST (the recommended screening cut-point for epidemiological research)[33] were invited for a phase 2 diagnostic assessment, along with all children whose caregiver indicated on the CFR that they had been referred for any of the following reasons: language delay/disorder, attention deficit hyperactivity disorder/attention deficit disorder, dyspraxia, autism spectrum condition (autism), genetic/chromosomal abnormality, epilepsy, Tourette syndrome and tuberous sclerosis. There was also a space on the CFR for the caregiver to provide more information about the reason for the referral and whether any diagnosis had been made. If the caregiver indicated that their child had been referred for any other reason not specified on the CFR, the research team decided on a case-by-case basis whether or not to invite the family for diagnostic assessment. All children who participated in phase 1 diagnostic assessments were invited for a phase 2 diagnostic assessment in order to assess the stability of the diagnostic outcome from phase 1 to phase 2.

### Diagnostic outcome

In both phase 1 and phase 2, diagnostic outcome was defined by the researchers reaching a consensus diagnosis using all diagnostic assessment data and researcher judgement as possible autism or autism spectrum (if they met the International Statistical Classification of Diseases 10th Edition criteria[41] for a pervasive developmental disorders diagnosis), atypical (if the child showed developmental concerns that were not related to autism) or typical (if there were no developmental concerns about the child). We used the term 'possible' to reflect the reluctance of some clinicians to commit to an autism diagnostic label at such an early age. It was not possible for all of the diagnostic assessment team to remain blind to the diagnostic outcome from the phase 1 diagnostic assessments at phase 2. However, the diagnostic assessment team re-examined blind to the Q-CHAT score throughout phase 2 diagnostic assessments.

The University of Cambridge was the sponsor of this research. Informed consent was obtained from all caregivers included in the study.

## Measures

### Quantitative Checklist for Autism in Toddlers

The Q-CHAT has a forced choice design, with five possible responses for each item (with the exception of item 4, see below). For each item, the response representing the most 'autistic' symptomatology scores 4 points and the least 'autistic' response scores 0. On approximately half the items (items 3, 7, 8, 11, 12, 13, 16, 18, 20, 22, 23, 24, 25) the 'autistic' response is the positive one. On the other half (items 1, 2, 4, 5, 6, 9, 10, 14, 15, 17, 19, 21) the 'autistic' response is the negative one. Item 4 is concerned with speech and includes a sixth option, 'my child does not speak', also scoring 4 points. The total Q-CHAT score is obtained by summing the score on each item, giving a maximum of 100. Each of the 25 questions is accompanied by a colour illustration to attempt to increase both response and comprehensibility of the items.

### Demographic data

Data collected about the child included age, sex, birth order, ethnicity and multiple birth status. Data collected about the caregivers included socioeconomic status (SES) and parental educational attainment. SES was derived at person level rather than household level, using the five class system of the National Statistics Socio-economic Classification.[42] Parental educational attainment was assessed by the highest level of qualification of each parent.

### Diagnostic assessment

The Autism Diagnostic Observation Schedule-Generic (ADOS-G)[43] was completed. The ADOS-G is a semistructured, standardised assessment of social interaction, communication, play and imagination. It is a reliable and valid measure for the diagnosis of autism relative to those with non-autistic conditions. Module 1 was chosen at phase 1 since it was not expected that many children would have expressive language abilities beyond basic phrase speech. Module 2 or 3 was chosen at phase 2. ADOS-G assessments were videotaped, but item scoring was conducted independently by two members of the diagnostic assessment team and consensus codes were agreed as soon as possible after the diagnostic assessment. The diagnostic assessment team comprised at least one very experienced research psychologist, alongside a research assistant who also had been trained to reliability standards.

The Autism Diagnostic Interview-Revised (ADI-R)[44] was completed with the caregiver. The ADI-R is a diagnostic semistructured interview that generates an algorithm score based on behaviours in three domains: social communication, social interaction and repetitive and stereotyped behaviours and interests. The interview also enquires about age of onset of symptoms. Due to the practicalities in arranging the diagnostic assessments, it was not possible to counterbalance completion of the ADOS-G and ADI-R. However, the ADI-R and ADOS-G were never carried out by the same researcher to reduce the possibility of experimenter expectation bias. All ADI-R assessments were audio-taped.

The Mullen Scales of Early Learning (MSEL)[45] is a standardised measure of cognitive, expressive and receptive language, motor and perceptual abilities. The MSEL was usually carried out while the other researcher was completing the ADI-R with the caregiver. The gross motor scale was not completed as this does not contribute to the overall early learning composite.

The Vineland Adaptive Behavior Scale (VABS)[46] is a standardised measure of personal and social skills. The interview version was completed with the parent. There are four domains that measure communication, daily living skills, socialisation and motor skills, as well as an optional maladaptive domain (not completed here). The VABS was usually completed at the end of the testing session.

## Statistical methods

Due to the stratified sample all analyses were adjusted for the probability of being selected into the diagnostic assessment subsample at each phase via inverse probability weighting. This takes account of the differences in sampling proportions and the selective participation in the diagnostic assessment across the sampling groups. A sensitivity analysis was undertaken that also included the non-response rates at each health trust within the weights. A receiver operating characteristic (ROC) area under the curve analysis assessed the discriminant power of the Q-CHAT and CAST and CFR in distinguishing autism cases from non-autism cases (defined as typical and atypical) and from autism and all other atypical development from typically developing children, across all score values. CIs were obtained from robust SEs. All analyses were undertaken in Stata V.14.[47] A series of sensitivity analyses to decisions on the screening instrument were undertaken (see table 2). At both phases, these included examining cut-points at 31 and 38, as well as a cut-point at 39, including those parents who reported concerns, and 39 adjusted for initial non-response at screening.

## RESULTS
### Phase 1

There were 3770 Q-CHAT questionnaires returned: 436 (14% response) from Luton, 1256 (25% response) from Bedfordshire and 2078 (42% response) from Cambridgeshire. Of these 223 were selected for diagnostic assessment, 32 did not give consent to be contacted and therefore 191 were invited and 121 completed the diagnostic assessments. See table 1 for phase 1 sampling strategy with study numbers, online supplemental figure 1 for the flow chart from phase 1, and online supplemental table 1 for the characteristics of children and their families throughout the study; online supplemental figure 2 shows the distribution of the Q-CHAT with sampling bands indicated.

Following diagnostic assessment, 11 children were defined as possible autism, 16 as atypical (7 language delay, 5 developmental delay, 4 other atypical) and 94 as typical. At a cut-point of ≥39, the PPV for autism was 17% (95% CI 8% to 31%) (table 2).

### Phase 2

From phase 1 caregivers, 3472 consenting caregivers were sent the phase 2 study materials, and 2005 returned both the CFR and the CAST (58% response), a further 6 returned only the CAST, and 5 returned only the CFR. Forty-two children who had a diagnostic assessment in phase 1 did not return the CAST or the CFR.

In phase 2, 172 caregivers reported developmental concerns on the CFR and 43 children scored 15 and above on the CAST (not in phase 1 diagnostic assessment sample) and were selected for diagnostic assessment. Eighty-one children who had been assessed in phase 1 completed the diagnostic assessment (1 child did not complete all the diagnostic assessments and was therefore not included in the analysis). In total, 158 children completed the diagnostic assessment at phase 2, 23 with CFR concerns and CAST score ≥15, 63 with CFR concerns and CAST scores ≤15, 6 with CAST scores ≥15 with no CFR concerns, 56 who scored ≤15 on the CAST and who did not report any developmental concerns on the CFR but who were assessed in phase 1, and 10 children who were assessed in phase 1 but who did not return the CFR or CAST. Figure 1 summarises the overall study design, with figure 2 showing the characteristics and comparison

---

**Table 1** Phase 1 follow-up diagnostic assessment sampling strategy, with study numbers

| Sampling group | Observed Q-CHAT (missing items=0) | Maximum Q-CHAT (missing items=4) | % sampled | Total | Assessment numbers | | |
| | | | | | Selected | Invited | Agreed |
|---|---|---|---|---|---|---|---|
| 1 | ≥38 | ≥44 | 100 | 86 | 86 | 74 | 40 |
| 2 | ≤37 | ≥44 | 50 | 21 | 10 | 9 | 6 |
| 3 | 41–43 | 41–43 | 50 | 61 | 31 | 28 | 16 |
| 4 | ≤40 | 41–43 | 50 | 21 | 10 | 8 | 5 |
| 5 | 38–40 | 38–40 | 25 | 129 | 31 | 28 | 20 |
| 6 | ≤37 | 38–40 | 25 | 25 | 5 | 5 | 2 |
| 7 | ≤37 | ≤37 | 1 | 3300 | 40 | 39 | 32 |
| Total | | | | 3643 | 213 | 191 | 121 |

Q-CHAT, Quantitative Checklist for Autism in Toddlers.

**Table 2** Autism diagnosis results: phase 1 and phase 2

| Autism | TP | FN | FP | TN | Sensitivity | Specificity | PPV | NPV | AUC |
|---|---|---|---|---|---|---|---|---|---|
| Main analysis: phase 1 | | | | | | | | | |
| Q-CHAT 39+ | 36 | 0 | 179 | 3428 | 1.00 (0.72 to 1.0)* | 0.95 (0.92 to 0.97) | 0.17 (0.08 to 0.31) | 1.00 (0.93 to 1.0)* | 0.98 (0.96 to 1.00) |
| Main analysis: phase 2 | | | | | | | | | |
| Q-CHAT 39+ | 27 | 34 | 69 | 3513 | 0.44 (0.26 to 0.64) | 0.98 (0.97 to 0.99) | 0.28 (0.15 to 0.46) | 0.99 (0.98 to 0.99) | 0.86 (0.78 to 0.94) |
| CAST ≥15+/CFR positive | 50 | 7 | 98 | 3488 | 0.88 (0.63 to 0.97) | 0.97 (0.95 to 0.98) | 0.34 (0.23 to 0.46) | 1.00 (0.99 to 1.00) | 0.99 (0.98 to 1.00) |
| Sensitivity analysis: phase 1 | | | | | | | | | |
| Q-CHAT 31+ | 36 | 0 | 1104 | 2504 | 1.00 (0.72 to 1.0)* | 0.69 (0.53 to 0.82) | 0.03 (0.01 to 0.07) | 1.00 (0.88 to 1.0)* | 0.98 (0.96 to 1.00) |
| Q-CHAT 38+ | 36 | 0 | 238 | 3369 | 1.00 (0.71 to 1.0)* | 0.93 (0.90 to 0.96) | 0.13 (0.06 to 0.25) | 1.00 (0.92 to 1.0)* | 0.98 (0.96 to 1.00) |
| Q-CHAT 39+ with parental concern | 30 | 6 | 73 | 3534 | 0.84 (0.43 to 0.97) | 0.98 (0.97 to 0.99) | 0.29 (0.13 to 0.52) | 1.00 (0.99 to 1.00) | 1.00 (0.99 to 1.00) |
| Q-CHAT 39+ adjust initial non-response | 49 | 0 | 201 | 3393 | 1.00 (0.72 to 1.0)* | 0.94 (0.91 to 0.97) | 0.20 (0.09 to 0.37) | 1.00 (0.93 to 1.0)* | 0.98 (0.96 to 1.00) |
| Sensitivity analysis: phase 2 | | | | | | | | | |
| Q-CHAT 31+ | 48 | 13 | 589 | 2993 | 0.79 (0.64 to 0.89) | 0.84 (0.65 to 0.93) | 0.08 (0.03 to 0.19) | 1.00 (0.99 to 1.00) | 0.86 (0.78 to 0.94) |
| Q-CHAT 38+ | 35 | 26 | 92 | 3489 | 0.57 (0.39 to 0.74) | 0.97 (0.96 to 0.99) | 0.28 (0.16 to 0.43) | 0.99 (0.99 to 1.00) | 0.86 (0.78 to 0.94) |
| Q-CHAT 39+ with parental concern | 24 | 37 | 29 | 3553 | 0.40 (0.23 to 0.60) | 0.99 (0.98 to 1.00) | 0.46 (0.24 to 0.69) | 0.99 (0.98 to 0.99) | 0.90 (0.84 to 0.96) |
| Q-CHAT 39+ adjust initial non-response | 38 | 35 | 73 | 3497 | 0.51 (0.30 to 0.72) | 0.98 (0.96 to 0.99) | 0.34 (0.17 to 0.55) | 0.99 (0.98 to 0.99) | 0.89 (0.81 to 0.96) |
| CAST ≥15 | 32 | 25 | 8 | 3578 | 0.56 (0.37 to 0.74) | 1.00 (0.99 to 1.00) | 0.81 (0.59 to 0.92) | 0.99 (0.99 to 1.00) | 0.94 (0.88 to 0.99) |
| Q-CHAT 39+ includes parent-reported cases | 30 | 34 | 67 | 3512 | 0.47 (0.30 to 0.65) | 0.98 (0.97 to 0.99) | 0.31 (0.52 to 0.82) | 0.99 (0.98 to 0.99) | 0.86 (0.79 to 0.94) |

*One-sided CI due to perfect predictor (not weighted).

AUC, area under the receiver operating characteristic curve; CAST, Childhood Autism Spectrum Test; CFR, Checklist for Referral; FN, false negative cases; FP, false positive cases; NPV, negative predictive value; PPV, positive predictive value; Q-CHAT, Quantitative Checklist for Autism in Toddlers; TN, true negative cases; TP, true positive cases.

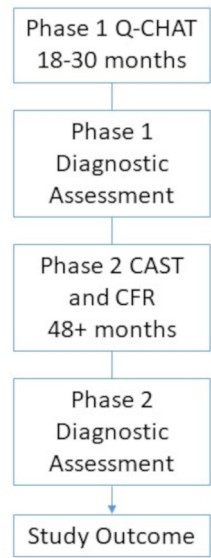

**Figure 1** Study design. CAST, Childhood Autism Spectrum Test; CFR, Checklist for Referral; Q-CHAT, Quantitative Checklist for Autism in Toddlers.

of the Q-CHAT at phases 1 and 2 and the CAST at phase 2 (not weighted). Eleven children were defined as possible autism in phase 1 (prevalence of 0.98%, 95% CI 0.45% to 2.16%).

Eighty-one children were assessed at both phase 1 and phase 2. Of the nine children whose diagnostic outcome at phase 1 was possible autism and who had a diagnostic assessment in phase 2, all were still classified with autism. A further six children were now classified with autism (four who were typical at phase 1 and two who were atypical). Seventy-seven children were assessed for the first time at

phase 2, having completed the Q-CHAT at phase 1. Some of these children were invited for diagnostic assessment due to indicating on the CFR that they had received a clinical diagnosis of autism and others because of their CAST score (15 or above). Of these 29 were classified as autism, 24 as atypical development and 24 as typically developing, giving a prevalence of autism of 1.57% (95% CI 0.90% to 2.74%). Figure 3 shows participants' study flow through screening and diagnostic assessments from phase 1 to phase 2. Clinical characterisation of the sample by screening status will be reported in a separate paper.

Using the CAST alone as the screen at phase 2, 1994 children had complete data. At a cut-point of ≥15, the sensitivity was 56% (95% CI 37% to 74%), specificity was 100% (95% CI 99% to 100%) and PPV was 81% (95% CI 59% to 92%) (table 2). The area under the ROC curve was 0.94 (95% CI 0.88 to 0.99) (online supplemental figure 3). Adding in the CFR selection criteria improved sensitivity to 88% (95% CI 63% to 97%), but decreased specificity slightly to 97% (95% CI 95% to 98%), with a PPV of 34% (95% CI 23% to 46%) and an area under the ROC curve of 0.99 (95% CI 0.98 to 1.00). Online supplemental figure 4 shows a box plot of the distribution of Q-CHAT scores split by both phase 1 and phase 2 diagnosis.

At a cut-point of ≥39, the sensitivity of the Q-CHAT in predicting phase 2 outcome (autism vs atypical and/or typical) was 44% (95% CI 26% to 64%), specificity was 98% (95% CI 97% to 99%) and PPV was 28% (95% CI 15% to 46%) (table 2). The area under the ROC curve was 0.86 (95% CI 0.78 to 0.94). The cut-point of 39 was determined by the data obtained at phase 1 and phase 2. We selected participants across the threshold of 44 following the Q-CHAT at phase 1 in varying proportions

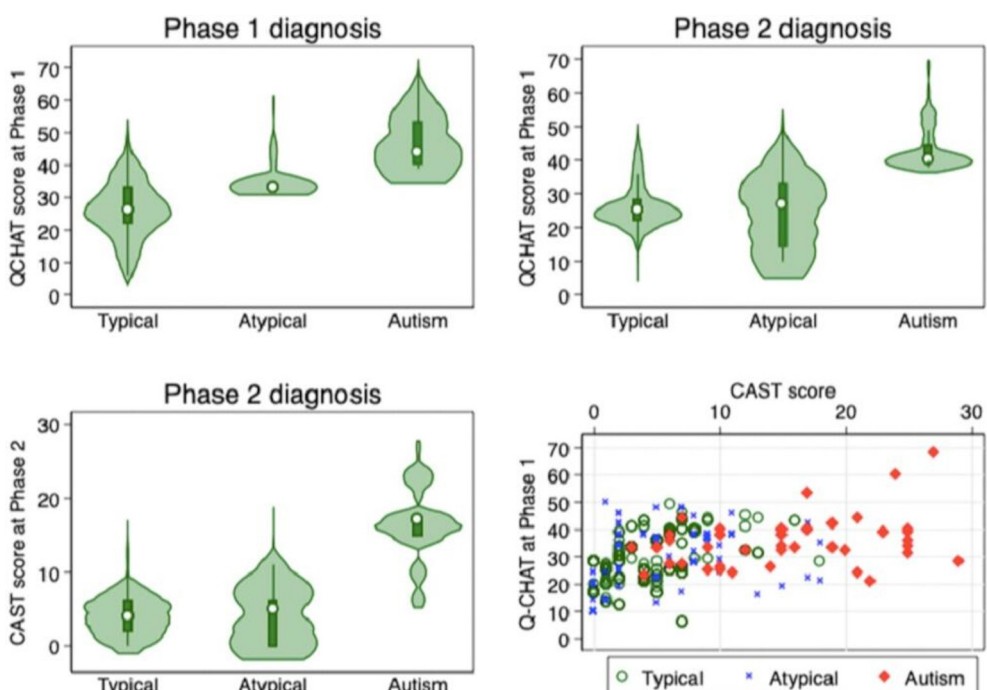

**Figure 2** Characteristics and comparison of the Q-CHAT at phase 1 and phase 2 and the CAST at phase 2 (not weighted). CAST, Childhood Autism Spectrum Test; Q-CHAT, Quantitative Checklist for Autism in Toddlers.

## PHASE 1

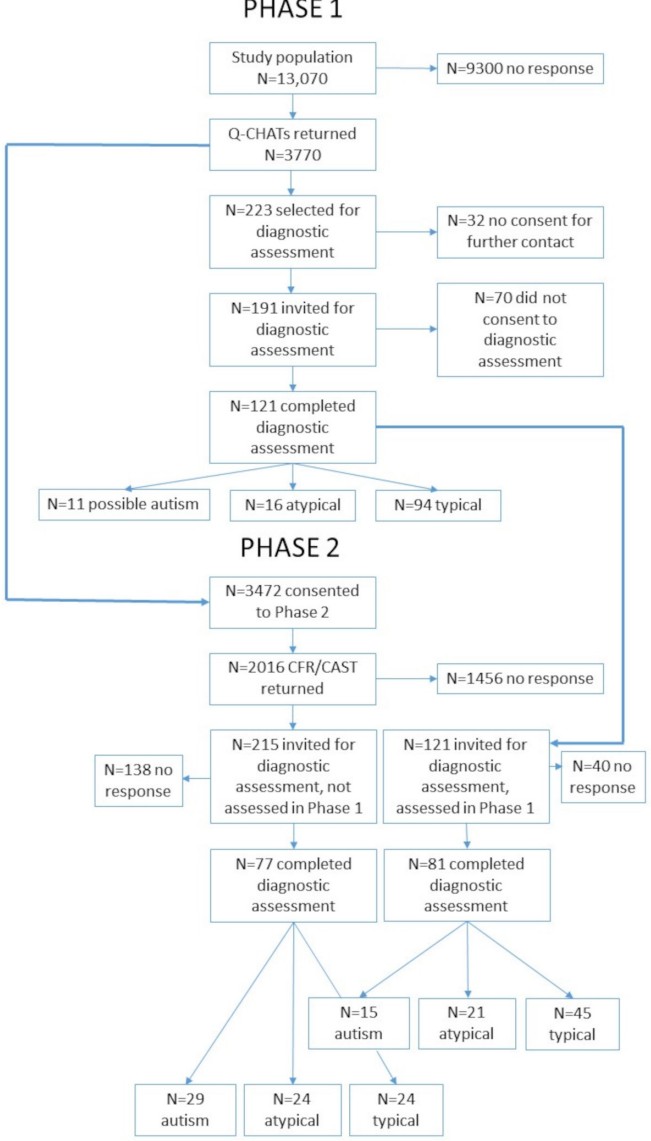

**Figure 3** Participant study flow through screening and diagnostic assessments from phase 1 to phase 2. CAST, Childhood Autism Spectrum Test; CFR, Checklist for Referral; Q-CHAT, Quantitative Checklist for Autism in Toddlers.

in order to accurately determine where the cut-point should lie. Examination of the Q-CHAT data against diagnostic outcome suggested that the cut-point of 39 is the point that maximises sensitivity and specificity at 18 to 30 months.

### Sensitivity analyses

Optimising the Q-CHAT cut-point for phase 2 outcome found two additional cut-points: Q-CHAT 31+ maximised sensitivity and specificity at the cost of false positives and using Q-CHAT 38+ marginally improved sensitivity with little additional benefit. Other sensitivity analyses at phase 1 and phase 2 are shown in table 2. See figure 2 for the characteristics and comparison of the Q-CHAT at phase 1 and phase 2 and the CAST at phase 2 (not weighted), according to diagnostic outcome.

## DISCUSSION

The first two aims of the study were to determine the optimal screening cut-point on the Q-CHAT and to determine the test accuracy. At phase 1 outcome, all children who were classified as possible autism failed the Q-CHAT at a cut-point of 39, demonstrating that early detection and diagnosis of autism is possible. The PPV of the Q-CHAT for autism was 17%. Misinterpretation of behaviours that are not well established as well as developmental immaturity may contribute to low PPV. An aim of phase 2 was to rescreen the population at a minimum age of 4 using the CAST in order to identify children with autism who were not identified at phase 1 (false negatives) and the CFR to seek information on the child's history of developmental concerns and diagnoses. The results indicated that the Q-CHAT did not identify all children who were later diagnosed with autism by age 4. We therefore were able to confirm the outcome of those who were identified by the Q-CHAT at 18–30 months and those who were not. A further aim in phase 2 was to confirm the discriminant power of the Q-CHAT and CAST and CFR in distinguishing autism cases from non-autism cases. At a cut-point of ≥39, the sensitivity of the Q-CHAT in predicting autism was 44% (95% CI 26% to 64%), specificity was 98% (95% CI 97% to 99%) and PPV was 28% (95% CI 15% to 46%). This demonstrates that the Q-CHAT alone cannot be used to identify all children with autism. This might reflect the Q-CHAT properties, but equally it likely reflects a subgroup of children with autism who do not show symptoms of sufficient 'severity' until later in childhood, and/or that symptoms of autism change with age, and so require developmentally sensitive instruments at different ages. The Diagnostic and Statistical Manual of Mental Disorders-5's caveat that symptoms might not fully manifest until social demands exceed capacities may apply here. The CAST picked up many of the cases of autism that were missed by the Q-CHAT, demonstrating the need for repeat screening rather than relying on a single instrument at one time-point. However, the sensitivity of the CAST at a cut-point of 15 was only moderate (56%), compared with our earlier study showing a sensitivity of 100%.[39] All children who participated in phase 2 diagnostic assessments who were identified as 'possible autism' at phase 1 were still classified as autistic, suggested the stability of the diagnosis from phase 1 to phase 2.

A strength of this study was the prospective, population-based design which allowed us to assess the Q-CHAT accurately at two time-points. Our approach to missing data and subsequent sampling strategy was conservative and ensured maximum capture of potential cases in the diagnostic assessment phase, maximising sensitivity. However, studies such as the current one are challenged by modest participation rates, selection bias and attrition over time. Despite adopting strategies to attempt to increase response, overall it was low, but not unusual for an unsolicited postal survey. Whether or not those who responded

initially to the Q-CHAT (which determined the population for the rest of the study) were biased towards those with concerns (either justified or the 'worried well') or in the opposite direction remains largely unknown. A second limitation is that in phase 1, only 1% of children who scored ≤37 were sampled. This proportion was chosen for pragmatic reasons in relation to feasibility (ie, to keep the number of lengthy diagnostic assessments in children with no developmental concerns to a minimum and to reduce expectation bias in the research team). None of the sampled children with scores ≤37 met our diagnostic assessment case definition. It is important to underline that these results are based purely on outcome at phase 1 and do not reflect the overall test accuracy statistics from phase 2.

Few studies attempt population screening partly because they are very time-consuming and expensive. Younger toddlers have mild or transient developmental delays that resolve, resulting in low PPV, as was found in another study.[48] Introducing a follow-up interview (such as used with the M-CHAT[25 27 28]) for those who screen positive may reduce the number of unnecessary referrals for a diagnostic assessment, especially with children older than 20 months.[49] Nevertheless, a recent meta-analysis of M-CHAT studies found a lack of evidence on its performance in low-risk children or at age 18 months.[50] A universal, primary care-based screening study was conducted using the M-CHAT/F in the USA. Overall, the sensitivity was 38.8% and the PPV was 14.6%. Sensitivity was higher in older toddlers and with repeated screenings, whereas PPV was lower in girls.[29] This study was the first using the M-CHAT to follow up children screened in the population, thus allowing for an accurate estimate of the sensitivity of the instrument. Similar results were found with another population-based sample (the Autism Birth Cohort), whereby the majority diagnosed with autism at 6-year follow-up scored below the cut-off on the M-CHAT at 18 months.[26] Since the publication of the original CHAT,[7 8 18] no study in the UK to date has achieved screening data on such a large population sample of toddlers and accurately determined sensitivity through follow-up.

Total Q-CHAT score was used to determine which sampling band each child fell into. A sum (total) score based on equal weighting of all items may not be the most appropriate method by which to quantify an individual's autistic traits. Other work by our group has shown that there are at least two items on the Q-CHAT whereby the prevalence of endorsement of the trait in the autistic direction is lower for toddlers with a diagnosis of autism, compared with population controls (items 8 and 18).[37] One study[51] found specific item clusters which gave high sensitivity and specificity values at 18 months (100% and 93.9%, respectively). Item response analysis will be conducted to determine whether every item equally contributes useful information about the underlying latent dimensions. Defining a threshold on each factor that can be measured quantitatively (continuous approach) may be a more appropriate scoring method. Furthermore, exploration of different age-specific scoring algorithms should be implemented to determine which variables carry the greatest relative importance to optimise screening results.[3 49]

Overall, this study demonstrates that (1) the Q-CHAT identified every child who received an autism diagnosis at phase 1 outcome and therefore it is possible to detect autism at 18–30 months; and (2) it is not possible to identify every child at a very young age who will later be diagnosed with autism. The Q-CHAT can be used to screen toddlers at 18–30 months for autism to enable accelerated referral into the diagnostic pathway. However, the PPV is low, thus potentially generating a high number of children who may not have autism, but who may still require a developmental assessment to determine whether they warrant a referral for intervention and support. Outcome data from phase 2 revealed many children with autism that the Q-CHAT did not identify. The autism spectrum is broad and not every child will be detectable as a young toddler. Therefore, when looking at the Q-CHAT longitudinally, it is evident that screening at a single time-point results in less than optimal test properties. In a future study, we will evaluate whether the missed cases reflect a subgroup of children with autism whose symptoms are not severe enough to warrant a diagnosis during toddlerhood and that the symptoms of autism change with age. We therefore recommend continued surveillance and rescreening at multiple time-points using developmentally sensitive instruments in order to identify a higher proportion of cases for maximal public health impact, which is in line with the recommendation by the American Academy of Pediatrics.[52] This study demonstrates that no single instrument will identify all cases of autism at a single time-point. Thus, continued monitoring throughout the life course is necessary in order to identify when autistic traits begin to impair an individual's life. High-quality evidence from random controlled trials is needed to determine whether early detection and consequent early implementation of specific interventions is able to change outcomes in toddlers with autism.

**Author affiliations**

[1]Psychiatry Department, Autism Research Centre, University of Cambridge, Cambridge, UK

[2]Population Health Sciences Institute, Campus for Ageing and Vitality, Newcastle University, Newcastle upon Tyne, UK

[3]Institute for Biomedical Research and Innovation (IRIB) - National Research Council of Italy (CNR), Messina, Italy

[4]Dept of Psychology, Institute of Psychiatry, Psychology & Neuroscience, King's College London, London, UK

[5]Cambridge Public Health, University of Cambridge, Cambridge, UK

**Acknowledgements** We are grateful to the families who participated in all phases of the study. We are also grateful to staff at Luton, Bedfordshire and Cambridgeshire CCGs for their cooperation in defining the target population and mailing the questionnaires. We are grateful to Gina Gomez, Kristelle Hudry, Sally

Clifford, Georgina Woods, Emma Robson, Philippa Lewington, Susan Sadek, Kimberly Armstrong, Martine Roelfsema and Kim Davies for help with data collection.

**Contributors** CA, FEM, CB, TC and SB-C contributed to study design. CA, LR, GP and RS collected the data. CA and FEM did the data analysis and FEM is the guarantor of the analysis. All authors contributed to data interpretation and writing of the report.

**Funding** The study was funded by the National Lottery Community Fund and the Autism Research Trust, and was conducted in association with the Gillings Family Charitable Trust and the National Institute for Health Research (NIHR) Collaboration for Leadership in Applied Health Research and Care (CLAHRC) East of England at Cambridgeshire and Peterborough NHS Foundation Trust, and by the NIHR Cambridge Biomedical Research Centre. The views expressed are those of the author(s) and not necessarily those of the NHS, the NIHR or the Department of Health. SB-C and TC were supported by the Medical Research Council UK and the Innovative Medicines Initiative (IMI) 2 Joint Undertaking (JU) under grant agreement number 777394. The JU receives support from the European Union's Horizon 2020 research and innovation programme and EFPIA and Autism Speaks, Autistica and SFARI. FEM was supported by MRC U105292687. SB-C and CA also received funding from the Wellcome Trust (214322\Z\18\Z), the Templeton World Charitable Foundation and SFARI. For the purpose of open access, SB-C has applied a CC BY public copyright licence to any author accepted manuscript version arising from this submission. The funding source had no role in the data collection, data analysis, data interpretation or writing of the report. The corresponding author had full access to all the data in the study and had final responsibility for the decision to submit for publication.

**Competing interests** TC has received research grant support from the Medical Research Council (UK), the National Institute for Health Research, Horizon 2020 and the Innovative Medicines Initiative (European Commission), MQ, Autistica, FP7 (European Commission), the Charles Hawkins Fund and the Waterloo Foundation. He has served as a consultant to F Hoffmann-La Roche. He has received royalties from Sage Publications and Guilford Publications. All other authors report no biomedical financial interests or potential conflicts of interest.

**Patient consent for publication** Not required.

**Ethics approval** Ethical approval was granted by the Cambridge local research ethics committee (05/Q0108/53).

**Provenance and peer review** Not commissioned; externally peer reviewed.

**Data availability statement** No data are available. At the time of data collection, consent was not obtained from participants for the purposes of data sharing. Therefore even anonymised data are not available.

**ORCID iDs**
Carrie Allison http://orcid.org/0000-0003-2272-2090
Fiona E Matthews http://orcid.org/0000-0002-1728-2388

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
