## [Reviewer comments · BMJ Paediatrics Open]

ARTICLE DETAILS

TITLE (PROVISIONAL)	The Quantitative Checklist for Autism in Toddlers (Q-CHAT). A population screening study with follow-up: the case for multiple time-point screening for autism.
AUTHORS	Allison, Carrie Matthews, Fiona E. Ruta, Liliana Pasco, Greg Soufer, Renee Brayne, Carol Charman, Tony Baron-Cohen, Simon

VERSION 1 – REVIEW

REVIEWER	Reviewer name: Dr. Serena Petrocchi Institution and Country: Not applicable Competing interests: None
REVIEW RETURNED	05-Oct-2020

GENERAL COMMENTS	Dear Editor and Authors, many thanks for the possibility to revise this paper. The study reported findings regarding a screening procedure carried out in the U.K. on a large sample of toddlers aged 18-30 months. The study examined the accuracy of the Q-CHAT in a cohort of children screened at 18-30 months and followed-up at age 4.. I have several major and minor points to suggest. Page 4 - Line 7: in the what is already known on the topic section, the authors should clarify whether in the U.K. is not recommend a "universal screening procedure" or a "universal screening tool". Furthermore, the authors stated that there is a limited research evidence on the longitudinal application of screening tools. Did the authors refer to study carried out only in the U.K.? If not, I suggested taking into account some reviews recently published. For example: Marlow, M.; Servili, C.; Tomlinson, M. A review of screening tools for the identification of autism spectrum disorders and developmental delay in infants and young children: Recommendations for use in low- and middle-income countries. Autism Res. 2019, 12, 176–199. Sánchez-García, A.B.; Galindo-Villardón, P.; Nieto-Librero, A.B.; Martín-Rodero, H.; Robins, D.L. Toddler Screening for Autism Spectrum Disorder: A Meta-Analysis of Diagnostic Accuracy. JADD 2019, 49, 1837–1852. Thabtah, F.; Peebles, D. Early Autism Screening: A Comprehensive Review. Int. J. Environ. Res. Public Health 2019, 16, 3502. Petrocchi, S., Levante, A., & Lecciso, F. (2020). Systematic Review of Level 1 and Level 2 Screening Tools for Autism Spectrum Disorders in Toddlers. Brain sciences, 10(3), 180. Introduction section:
---

	Page 5 - Line 5: Please, upgrade the references regarding the prevalence. The authors could refer to the CDC website and/or to the papers on epidemiological data. For example: Maenner, M.J., Shaw, K.A., Baio, J., Washington, A., Patrick, M., DiRienzo, M., et al. (2020). Prevalence of Autism Spectrum Disorder Among Children Aged 8 Years — Autism and Developmental Disabilities Monitoring Network, 11 Sites, United States, 2016. Morbidity and mortality weekly report. Surveillance summaries, 69(4), 1–12. When the authors described the different versions of the CHAT, I suggest them to refer to the reviews mentioned above, to describe critically the CHAT revisions and their psychometric properties. Methods: The study purposes were clearly reported; the procedure was detailed. The diagnostic battery included adequate comparative instruments. Statistical analyses and software applied were adequate for the study purposes. Page 7 - Insert the full stop in Patients and Public Involvement statement section. Page 8 - Line 7: The authors affirmed that the screening questionnaire was re-administered before to the diagnostic assessment. The authors should explain what procedure they applied whether the questionnaire scores decrease. Was the child still evaluated? If the parents changed the answers, did the authors examine which items have been contributed to the score? Page 8 - Lines 21-24: The cut-point was fixed to 44 according to two criteria. The authors should specify also which is the percentile corresponding to the threshold. Results: Page 13: The authors reported the diagnostic outcomes, but not the mean scores reached by children to the gold standard measures. Were there significant differences between children with autism and atypically developing children in those mean scores? I would suggest to report this information and the comparative analyses. Furthermore, the authors diagnosed 4 children with other atypically, please specified what are the scores in the diagnostic measures for those children. Discussion: Page 17 - Line 5: There is a double full stop. Page 18/19: When the authors discussed about the Q-CHAT total score, they could refer also to other studies. For example: Devescovi, R., Monasta, L., Bin, M., Bresciani, G., Mancini, A., Carrozzi, M., & Colombi, C. (2020). A Two-Stage Screening Approach with I-TC and Q-CHAT to Identify Toddlers at Risk for Autism Spectrum Disorder within the Italian Public Health System. Brain Sciences, 10(3), 184. Lecciso, F., Levante, A., Signore, F., & Petrocchi, S. (2019). Preliminary evidence of the Structural Validity and measurement invariance of the Quantitative-CHecklist for Autism in Toddler (Q-CHAT) on Italian unselected children. Electronic Journal of Applied Statistical Analysis, 12(2), 320-340. Supplementary materials: Regarding the tables, when you reported the acronym GCSE please add a note to explain better and/or refer to the EQF (European qualification framework).
--	---

REVIEWER	Reviewer name: Dr. Lonnie Zwaigenbaum
-----------------	---------------------------------------

	Institution and Country: University of Alberta Faculty of Medicine and Dentistry, Pediatrics, Canada Competing interests: None
REVIEW RETURNED	22-Oct-2020

GENERAL COMMENTS	This manuscript summarizes a multistage autism screening study. In 'Phase 1', 3770 toddlers were screened at 18-30 months using the Q-CHAT. This represented 29% of total sample of 13070 toddlers from a Primary Care Trust (child health surveillance database), invited by mail with one reminder. Stratified sampling based on Q-CHAT scores (and maximum scores accounting for incomplete responses/missing items) was used to select toddlers for follow-up. Of 191 invited, 121 completed diagnostic assessments (63%), of whom 11 were 'defined as autistic' (p 12, line 52), although described in Figure 1 as 'possible ASC'. In 'Phase 2', when the children were age 4, caregivers who consented to further contact (n=3472) were invited to complete two additional screening measures, the Childhood Autism Spectrum Test (CAST), and the Checklist for Referral (CFR). Of these, 2005 participated (58%). Children who were screen positive on either of these 4-year measures, or who participated in Phase 1 assessments, were offered a diagnostic assessment. I believe that 253 (172 who were screen positive at age 4, and 81 children who were assessed in Phase 1) were offered assessment, completed by 158 (62%). There were 29 new ASD diagnoses among this group, plus all 9 of the original 11 who were diagnosed at Phase 1 and seen at age four (2 lost to follow-up). The discussion focuses on the low PPV of the Q-CHAT at 18-30 months, the detection of cases on subsequent screening, and implications for the need for 'continued surveillance and re-screening at multiple points in time' given the heterogeneity of ASD. The manuscript has several strengths, including large, community-based sample, multi-point screening, and selected follow-up of children that included those who were screen positive and screen negative. The manuscript is generally clear and well-written (although was difficult to track children through the various steps of the study in places; see below), and the discussion points, particularly the finding that no single early detection strategy is likely to identify all children with autism, have important implications for practice and policy. As the authors comment, a longitudinal community screening study is inherently complex and challenged by 'modest participation rates, selection bias and attrition' (pg 16, line 22-24). That said, further clarity within the text and figures would help the reader understand this rich dataset. There are several relatively minor points throughout the paper, summarized below for consideration. Introduction  - Guthrie et al. (2019) (ref 45), first mentioned in the Discussion, should be referenced when describing the M-CHAT in the Introduction, as it provides valid sensitivity estimates since ASD diagnoses were ascertained independent of the screen. It should also be acknowledged that higher participation rates were achieved than in the current study by incorporating ASD screening into routine check-ups, integrated into the electronic health record. - More detail about the findings of Allison et al (2008) should be provided; in particular, how these findings informed the current study, and what was the sample overlap. The Methods refer to a cut-point of 44 identifying 70% of children with ASD, so presumably this initial study included some follow-up and diagnostic assessments. Similarly, how was the cut-point of 39 used in the present study determined? Methods  - Should clarify what constituted a positive screen on the Q-CHAT. Observed and maximum (ie taking account of potentially higher scores based on missing items) were calculated. I can understand the rationale in relation to selecting children for follow-up but could the
--

	child be 'screen positive' (i.e., a score of 39 or higher) based on either the observed or maximum score?  - For Phase 2, clarify the rationale of inviting those children who 'participated in Phase 1 assessment', but not children who were actually screen positive on the Q-CHAT, even if they didn't participate. Similarly, clarify how the sampling approach that determined which children were seen for Phase 2 assessments provided valid estimates of the sensitivity and specificity of the Q-CHAT relative to 4-year diagnoses? Results  - I don't think that Figure 3 is referenced in the text. It is difficult to follow. It appears the intention is to describe subject flow through Phase 2, but it might be better to map the flow through the study as a whole. The blue font (numbers) is also difficult to read. The 15 cases of ASD noted under Phase 1- is this the 9 initially identified from Phase 1 assessment, plus 6 from Phase 2 assessment from among those invited back for assessment just because they participants in Phase 1 assessment? It took a while to figure that out, although in fairness is noted in the text. - A rationale for the various sensitivity analyses is also needed (see page 12, line 21-22). I can see the value, but it is also challenging to work through the classification data in Table 2. Could further elaboration be added to the appended materials? - As noted in the opening paragraph of this review, of the children assessed at Phase 1, 11 were 'defined as autistic' (p 12, line 52), although described in Figure 1 as 'possible ASC'. Was this distinction intended, and why is ASC being qualified with 'possible'? - Also, page 14, line 31, refers to 11 children were considered as 'possibly autistic' (prevalence of 0.98%, 95%CI 0.45%-2.16%), although in reference to being assessed at both Phase 1 and 2 (I think? This paragraph was somewhat confusing and was difficult to map onto Figure 2) Regardless, what denominator was used to calculate 0.98%? - Given the nature of the 2nd screening text at age 4 (CFR = Checklist for Referral), one might wonder if some of the participants were clinically diagnosed with ASD between the Phase 1 and 2 assessments. Was that the case? Discussion  - Page 16, first paragraph - suggests that children with milder symptom severity (or those whose symptoms increase over time) might not be detected until later in childhood. The authors have data on this point - the ADOS scores from the Phase 1 and 2 assessments. Similarly the authors list the MSEL and Vineland in the measures section, but do not report data. Providing some descriptive data on the early vs. later detected cases would add important nuance to the manuscript and would avoid the need to speculate on what was the nature of these differences. - Page 17, line 16. The authors conclude, 'it is possible to detect autism at 18-30 months'. It would help to describe the age distribution of these early diagnosed cases (total n=11; for example, how many were younger than 2 years?) - Although the authors may have been tight for space, consider ending the manuscript on a less technical note (e.g., something about potential application of the Q-CHAT)? Overall, the authors are to be commended for completing such an ambitious study, which does offer insights about the challenges of community-based ASD screening and the importance of ongoing surveillance and follow-up.
--	--

REVIEWER	Reviewer name: Dr. Eirini Koutoumanou Institution and Country: University College London, United Kingdom of Great Britain and Northern Ireland Competing interests: None
REVIEW RETURNED	31-Oct-2020

GENERAL COMMENTS

This report deals with a very important question and succeeds in answering its ultimate research goal. I have listed below some comments/suggestions that I would invite the authors to consider and provide their feedback on:

- Which statement in the Results section of the Abstract supports the following statement: "For every true positive there would, however, be 5 false positives." – could you please clarify for the readers' benefit?

- Could a reference please be added for the following statement "...reaches a minimum of 80% sensitivity and specificity" ?

- As the M-CHAT-R/F is described as the 'better' latest version of the CHAT, did the authors consider dimensionalising autism in toddlerhood via a Quantitative version of the M-CHAT-R/F instead of the original CHAT? Or would you say that this is in essence what was done ?

- Questionnaires were not sent exactly at the same time to all PCTs – why is that and are there any potentially (seasonal or other) effects/biases introduced from this move? If so, these should be acknowledged.

- Could you please clarify how the first sentence of the following statement leads on to the next one (I'm unclear that it does or it simple needs rephrasing to explain what 'at reduced frequency' means): "Each item is converted to a rating scale, thus quantifying autistic traits. This allows for the possibility of reporting behaviour at a reduced frequency."

- "Phase 1 was not evaluated at the time to ensure researchers remained blind to the results when undertaking Phase 2." – do the authors mean that diagnosis was not evaluated at the time for Phase 1? I don't think it's clear at it stands.

- Could you please clarify what it's meant by the following: "The Q-CHAT was re-administered prior to any of the assessment battery."?

- The authors employed a very conservative approach for dealing with missing data, however I am not sure how this was later incorporated in the analysis. In other words, how was movement across score groups from observed and maximum score actually taken into account (as implied in the text – end of Phase 1 Sampling strategy section)?

- Do the authors believe there is scope for analysing the Q-CHAT scores in their numerical format as opposed to categories? It might worth briefly saying something about this if you think appropriate (I would certainly like to know your thoughts).

- SF1 and numbers at the start of Phase 1 Results section do not match up: 223 selected and 32 not consented vs 213 and 22 respectively in SF1.

- It's not clear to me what the authors mean by "counterbalance administration of the ADOS-G and ADI-R."

- The type of graphs used in Figure 3 are not generally very commonly seen and readers might struggle to read them efficiently. Could these be turned into (or supplemented by) line diagrams across the different phases for those that were assessed at both phases? I.e. phase 1 and 2 on the x axis, different lines for typical, atypical and autism and two y axis, one for QCHAT score at phase 1 and another for phase 2?

- The final paragraph feels like an odd way to conclude this paper. Its content is useful and interesting, but I believe it would fit better earlier in the Discussion section to give space for the concluding remarks of the paper that relate directly to the original research

	question. Minor correction/edits:  - Please expand the "DSM-III-R" acronym in the Introduction section - "Luton and Bedfordshire, and Cambridgeshire" -> changed to "Luton, Bedfordshire and Cambridgeshire" or is "Luton and Bedfordshire" the preferred official term of this county/area? - I recommend "...at the specified Primary Care Trusts..." is replaced by "...at 3 Primary Care Trusts: Luton, Bedfordshire and Cambridgeshire..." - Replace "...and the sampling strategy was applied..." by "...and the sampling strategy as detailed below, was applied..." - What does random.org 8 stand for/mean? - "...questionnaires were returned: 436 from Luton..." – notice the newly added colon - Add a footnote on table 2 to explain the acronyms, TP, FN, FP, TN, PPV, NPV, ROC for the readers' benefit. Similarly, for Figure 1 to explain ASC. - Finally, I personally found the use of the word administer/administration when referring to the use of questionnaires/interviews, etc a bit odd but I appreciate this is very subjective.
--	--

VERSION 1 – AUTHOR RESPONSE

Re: Manuscript ID bmjpo-2020-000700 - "The Quantitative Checklist for Autism in Toddlers (Q-CHAT). A population screening study with follow-up: the case for multiple time–point screening for autism."

Thank you very much for providing us the opportunity to make these revisions. We are grateful for the careful review that the editors and reviewers have given to our paper. The table below summarises how we have addressed the comments/reviews in the manuscript.

	Reviewer comments	Amendments
	Reviewer #1	
1	Page 4 - Line 7: in the what is already known on the topic section, the authors should clarify whether in the U.K. is not recommend a "universal screening procedure" or a "universal screening tool".	For clarity, this has been amended to: In the UK, a systematic population screening programme for autism is not recommended to facilitate early detection, because general population screening tests that have been evaluated using systematic follow-up have not proved accurate.
2	Furthermore, the authors stated that there is a limited research evidence on the longitudinal application of screening tools. Did the authors refer to study carried out only in the U.K.? If not, I suggested taking into account some reviews recently published. For example: Marlow, M.; Servili, C.; Tomlinson, M. A review of screening tools for the identification of autism spectrum disorders and developmental delay in infants and young children: Recommendations for use	Thank you for sharing these recent reviews. Due to the limited space in the 'What is already known on this topic' section, this has been re-worded to follow up further on the original point, which is that screening is not recommended in the UK. It now reads: In the UK, a systematic population screening programme for autism is not recommended to facilitate early detection, because general population screening tests that have been evaluated using systematic follow-up have not proved accurate.

	Reviewer comments	Amendments
	in low-and middle-income countries. Autism Res. 2019, 12, 176–199. Sánchez-García, A.B.; Galindo-Villardón, P.; Nieto-Librero, A.B.; Martín-Rodero, H.; Robins, D.L. Toddler Screening for Autism Spectrum Disorder: A Meta-Analysis of Diagnostic Accuracy. JADD 2019, 49, 1837–1852. Thabtah, F.; Peebles, D. Early Autism Screening: A Comprehensive Review. Int. J. Environ. Res. Public Health 2019, 16, 3502. Petrocchi, S., Levante, A., & Lecciso, F. (2020). Systematic Review of Level 1 and Level 2 Screening Tools for Autism Spectrum Disorders in Toddlers. Brain sciences, 10(3), 180.	We have cited one of these reviews in the main manuscript (although these reviews do not include follow-up of the screened populations/samples).
3	Page 5 - Line 5: Please, upgrade the references regarding the prevalence. The authors could refer to the CDC website and/or to the papers on epidemiological data. For example: Maenner, M.J., Shaw, K.A., Baio, J., Washington, A., Patrick, M., DiRienzo, M., et al. (2020). Prevalence of Autism Spectrum Disorder Among Children Aged 8 Years — Autism and Developmental Disabilities Monitoring Network, 11 Sites, United States, 2016. Morbidity and mortality weekly report. Surveillance summaries, 69(4), 1–12.	We have added this reference, thank you.
4	When the authors described the different versions of the CHAT, I suggest them to refer to the reviews mentioned above, to describe critically the CHAT revisions and their psychometric properties.	Thank you for this suggestion, I have directed the reader to the Petrocchi et al (2020) systematic review.
5	Page 7 - Insert the full stop in Patients and Public Involvement statement section.	This has been amended, thank you.
6	Page 8 - Line 7: The authors affirmed that the screening questionnaire was re-administered before to the diagnostic assessment. The authors should explain what procedure they applied whether the questionnaire scores decrease. Was the	Thank you for highlighting this was not clear in the manuscript. The assessment team were completely blind to the child's Q-CHAT score and the Q-CHAT was re-administered purely to get a sample to examine test retest reliability in a sample enriched with high scorers. The test result

	Reviewer comments	Amendments
	child still evaluated? If the parents changed the answers, did the authors examine which items have been contributed to the score?	had no influence on the assessment at all. The text has been amended as follows: The Q-CHAT was completed a second time prior to any of the assessment battery in order to later examine the test-retest reliability of the Q-CHAT in a sample enriched with high scorers, which will be reported separately.
7	Page 8 - Lines 21-24: The cut-point was fixed to 44 according to two criteria. The authors should specify also which is the percentile corresponding to the threshold.	The cutpoint of 44 represents 2.4% of the population.
8	Page 13: The authors reported the diagnostic outcomes, but not the mean scores reached by children to the gold standard measures. Were there significant differences between children with autism and atypically developing children in those mean scores? I would suggest to report this information and the comparative analyses. Furthermore, the authors diagnosed 4 children with other atypically, please specified what are the scores in the diagnostic measures for those children.	We thank the reviewer for this comment. We chose not to include any data from the diagnostic measures in this manuscript as we purely wanted to communicate the test accuracy result of the Q-CHAT itself. Adding in the data from the diagnostic assessments would increase the length and detract from the purpose of this paper. However, these will be reported in a separate paper focusing on the characterisations of the measures obtained at assessment. We have added a sentence in the manuscript on page 14 as follows: Clinical characterisation of the sample by screening status will be reported in a separate paper.
9	Page 17 - Line 5: There is a double full stop.	Thank you, this has been corrected.
10	Page 18/19: When the authors discussed about the Q-CHAT total score, they could refer also to other studies. For example: Devescovi, R., Monasta, L., Bin, M., Bresciani, G., Mancini, A., Carrozzi, M., & Colombi, C. (2020). A Two-Stage Screening Approach with I-TC and Q-CHAT to Identify Toddlers at Risk for Autism Spectrum Disorder within the Italian Public Health System. Brain Sciences , 10(3), 184. Lecciso, F., Levante, A., Signore, F., & Petrocchi, S. (2019). Preliminary evidence of the Structural Validity and measurement invariance of the Quantitative-CHecklist for Autism in Toddler (Q-CHAT) on Italian unselected children. Electronic Journal of Applied Statistical Analysis , 12(2), 320-340.	Thank you. We have included a reference to the Devescovi et al study as an example. The additional sentence is: One study ⁴⁸ found specific item clusters which gave high sensitivity and specificity values at 18 months (100% and 93.9 respectively).

	Reviewer comments	Amendments
11	Regarding the tables, when you reported the acronym GCSE please add a note to explain better and/or refer to the EQF (European qualification framework).	Thank you, a note has been added in the table.

	Reviewer #2	
1	Guthrie et al. (2019) (ref 45), first mentioned in the Discussion, should be referenced when describing the M-CHAT in the Introduction, as it provides valid sensitivity estimates since ASD diagnoses were ascertained independent of the screen. It should also be acknowledged that higher participation rates were achieved than in the current study by incorporating ASD screening into routine check-ups, integrated into the electronic health record.	Thank you for highlighting that this important study was missed from the introduction. An additional paragraph has been added: An additional study examined the accuracy of the M-CHAT/F in a universal, primary care-based screening context that involved systematic follow-up up to 8 years.²⁸ Diagnoses were ascertained independent of the screen. The study reported sensitivity was 38.8%, and positive predictive value (PPV) was 14.6%, with almost universal screening being achieved (91%) by incorporating autism screening into routine check-ups, and integrating these within the electronic health record.
2	More detail about the findings of Allison et al (2008) should be provided; in particular, how these findings informed the current study, and what was the sample overlap. The Methods refer to a cut-point of 44 identifying 70% of children with ASD, so presumably this initial study included some follow-up and diagnostic assessments. Similarly, how was the cut-point of 39 used in the present study determined?	Thank you for raising this point. There was no sample overlap, the preliminary study and the current study are independent. Unfortunately we were unable to collect systematic follow up data from the 2008 study due to resource limitations. We used 44 as a sampling cut-point in the current study due to the fact that over 70% of children who already had a diagnosis of autism from the 2008 study scored above 44 on the Q-CHAT. However, the eventual cut-point of 39 was determined by the data obtained in the current study. We sampled across the threshold of 44 in varying proportions in order to accurately determine where the cut-point should lie. It was only at the end of the study when analysing the Q-CHAT data against diagnostic outcome did it become clear that the cut-point of 39 is the point that maximises sensitivity and specificity at 18 months. The paragraph that introduces the Q-CHAT has been re-written as follows to make this clearer: A preliminary study examined the distribution of the Q-CHAT in an unselected sample of 18-24

		month old toddlers and in a sample of children already diagnosed on the autism spectrum. Results indicated that the Q-CHAT discriminated well between young autistic children and unselected toddlers in the population at 18–24 months.²⁹ Test accuracy data were not collected in this sample due to resource limitations. In the methods section, we have rewritten the paragraph referring to the cut-point of 44 as follows: The rationale for choosing 44 as the sampling cut-point was based on anticipated estimates of prevalence of autism (approximately 1%) balanced with the knowledge that over 70% of the children who already had a diagnosis on the autism spectrum scored above 44 in our initial study²⁹, maximising sensitivity whilst generating a feasible number of assessments to be completed. The eventual cut-point was not determined until analysis of Q-CHAT scores against diagnostic outcome was complete.
3	Should clarify what constituted a positive screen on the Q-CHAT. Observed and maximum (ie taking account of potentially higher scores based on missing items) were calculated. I can understand the rationale in relation to selecting children for follow-up but could the child be 'screen positive' (i.e., a score of 39 or higher) based on either the observed or maximum score?	Thank you for pointing out that this is not clear. It is important to remember that the cut-point of 39 was not used to determine a screen positive, since one of the aims of the study was to determine the most appropriate cut-point on the Q-CHAT. This is what makes this study slightly different from other studies examining the test accuracy of autism screening measures – we sampled across the score band that we thought the cut point might lie (based on our Allison et al 2008 study), and the cut-point was determined at the end of the study. Table 1 provides the Phase 1 sampling strategy, with study numbers. We have added an example illustrating the sampling strategy for clarity, as follows: The final sampling groups were determined according to the maximum score, taking into account the movement across score groups from observed score to maximum score. For example, if a child's observed score was 40 and there were two missing Q-CHAT items, eight was added to the observed score to give 48. Forty eight would therefore be that child's sampling score. Sampling group allocation

		and randomisation took place prior to establishing whether or not the family had consented for further contact.
4	For Phase 2, clarify the rationale of inviting those children who 'participated in Phase 1 assessment', but not children who were actually screen positive on the Q-CHAT, even if they didn't participate.	Thank you for raising this point. At the time that the Phase 2 assessments were conducted, the cut-point of 39 was unknown. We invited all the children who took part in Phase 1 to ensure we had captured all the possible cases within this sub-sample and assess the stability of the diagnostic outcome from Phase 1 to Phase 2, and hoped that any additional 'missed' cases from Phase 1 would be picked up by the Checklist for Referral and the CAST at Phase 2. An additional sentence has been added to clarify the purpose of inviting Phase 1 children for assessment as follows: All children who participated in Phase 1 assessments were invited for a Phase 2 assessment in order to assess the stability of the diagnostic outcome from Phase 1 to Phase 2.
5	Similarly, clarify how the sampling approach that determined which children were seen for Phase 2 assessments provided valid estimates of the sensitivity and specificity of the Q-CHAT relative to 4-year diagnoses?	The sampling approach was incorporated into the weights for each individual. All children that provided assessments were included in the analysis. All children who undertook either the screen or assessments were included in the analysis of the sensitivity and specificity.
6	I don't think that Figure 3 is referenced in the text.	Figure 3 is referenced in the text at the end of the third paragraph in the Phase 2 results section.
7	It [Figure 3] is difficult to follow. It appears the intention is to describe subject flow through Phase 2, but it might be better to map the flow through the study as a whole. The blue font (numbers) is also difficult to read. The 15 cases of ASD noted under Phase 1– is this the 9 initially identified from Phase 1 assessment, plus 6 from Phase 2 assessment from among those invited back for assessment just because they participants in Phase 1 assessment? It took a while to figure that out, although in fairness is noted in the text.	The flow of the whole study is complex to show in one figure. The assessment phase 1 and phase 2 numbers are shown in Figure 1. The 15 are shown in Figure 1 with the phase 1 outcomes (possible autism=9, atypical=7, typical=66). Figure 3 was only provided to give more detail on the longitudinal response to the screening interview but can be removed if felt to be unclear.
8	A rationale for the various sensitivity analyses is also needed (see page 12, line 21-	Sensitivity analysis were chosen to show the robustness impact on

	22). I can see the value, but it is also challenging to work through the classification data in Table 2. Could further elaboration be added to the appended materials?	the cutpoint selection (chosen as maximizing sensitivity / specificity at phase 1, then phase 2) plus the addition of other information that might assist the screening process. Additional information has been added to the supplementary information.
9	As noted in the opening paragraph of this review, of the children assessed at Phase 1, 11 were 'defined as autistic' (p 12, line 52), although described in Figure 1 as 'possible ASC'. Was this distinction intended, and why is ASC being qualified with 'possible'?	Apologies, the discrepancy in terminology has been corrected in both the text and in Figure 1. We qualified autism as possible at Phase 1 reflecting on the reluctance of some clinicians to commit to an autism diagnostic label at such an early age.
10	Also, page 14, line 31, refers to 11 children were considered as 'possibly autistic' (prevalence of 0.98%, 95%CI 0.45%–2.16%), although in reference to being assessed at both Phase 1 and 2 (I think? This paragraph was somewhat confusing and was difficult to map onto Figure 2) Regardless, what denominator was used to calculate 0.98%?	The denominator is the phase 1 study size with the number as considered possibly autistic weighted to take into account the sampling design.
11	Given the nature of the 2nd screening text at age 4 (CFR = Checklist for Referral), one might wonder if some of the participants were clinically diagnosed with ASD between the Phase 1 and 2 assessments. Was that the case?	Yes, this is the reason we used the CFR to identify children who received a diagnosis between the two time-points. We have added in a sentence to make it clearer as follows: 77 children were assessed for the first time at Phase 2. Some of these children were invited for assessment due to indicating on the CFR that they had received a clinical diagnosis of autism, and others because of their CAST score.
12	Page 16, first paragraph – suggests that children with milder symptom severity (or those whose symptoms increase over time) might not be detected until later in childhood. The authors have data on this point – the ADOS scores from the Phase 1 and 2 assessments. Similarly the authors list the MSEL and Vineland in the measures section, but do not report data. Providing some descriptive data on the early vs. later detected cases would add important nuance to the manuscript and would avoid the need to speculate on what was the nature of these differences.	We thank the reviewer for this comment. As mentioned earlier in relation to Reviewer 1's comment, we chose not to include any data from the diagnostic measures in this manuscript as we purely wanted to communicate the test accuracy result of the Q-CHAT itself. Adding in the data from the diagnostic assessments would increase the length and detract from the purpose of this paper. However, these will be reported in a separate paper focusing on the characterisations of the measures obtained at assessment. We have added a sentence as follows on page 14: Clinical characterisation of the sample by screening status will be reported in a separate paper.
13	Page 17, line 16. The authors conclude, 'it is possible to detect autism at 18-30 months'. It would help to describe the age	The screening was undertaken between 18-30 months to consider all cases within this age

	distribution of these early diagnosed cases (total n=11; for example, how many were younger than 2 years?)	range. We do not want to split the sample down into an arbitrary subgroup.
14	Although the authors may have been tight for space, consider ending the manuscript on a less technical note (e.g., something about potential application of the Q-CHAT)?	Thank you for this suggestion. We have restructured the discussion, ending with a reflection about the current study and its implications.
15	Overall, the authors are to be commended for completing such an ambitious study, which does offer insights about the challenges of community-based ASD screening and the importance of ongoing surveillance and follow-up.	We thank the reviewer for their kind comments.
	Reviewer #3	
1	Which statement in the Results section of the Abstract supports the following statement: “For every true positive there would, however, be 5 false positives.” – could you please clarify for the readers’ benefit?	Thank you for this suggestion. The statement refers to the low PPV at Phase 1 (The positive predictive value (PPV) at a cut–point of 39 was 17%). This has been clarified in the abstract as follows: The low PPV suggests that for every true positive there would, however, be 5 false positives.
2	Could a reference please be added for the following statement “...reaches a minimum of 80% sensitivity and specificity” ?	We thank the reviewer for highlighting that the reference to Glascoe (2015) was missing. This has been added.
3	As the M-CHAT-R/F is described as the ‘better’ latest version of the CHAT, did the authors consider dimensionalising autism in toddlerhood via a Quantitative version of the M-CHAT-R/F instead of the original CHAT? Or would you say that this is in essence what was done?	Thank you for this question. In fact, the Q-CHAT was designed at around the same time as the original M-CHAT, so the Q-CHAT remains a revision of the original CHAT, rather than any subsequent versions of the M-CHAT-R/F.
4	Questionnaires were not sent exactly at the same time to all PCTs – why is that and are there any potentially (seasonal or other) effects/biases introduced from this move? If so, these should be acknowledged.	Thank you for this question. The reasons for this was purely because of available resources of the team. We would not have had the capacity to screen and follow-up all three PCTs at the same time. It is possible that biases may have crept in as a result of this, this will be explored in a subsequent paper. An additional sentence has been added on page 7: Questionnaires were sent in three batches to manage capacity of the team.
5	Could you please clarify how the first sentence of the following statement leads on to the next one (I’m unclear that it does or it simple needs rephrasing to explain what ‘at reduced frequency’ means): “Each	Thank you for this query. We have added a sentence to clarify what is meant as follows: For example, the response options on the item concerning protodeclarative pointing, range

	item is converted to a rating scale, thus quantifying autistic traits. This allows for the possibility of reporting behaviour at a reduced frequency.”	from many times a day (least autistic response) through to never (most autistic response).
6	“Phase 1 was not evaluated at the time to ensure researchers remained blind to the results when undertaking Phase 2.” – do the authors mean that diagnosis was not evaluated at the time for Phase 1? I don’t think it’s clear at it stands.	Thank you for pointing out that this is not clear. We have added the following to hopefully clarify this point: Test accuracy at Phase 1 was not evaluated at the time to ensure researchers remained blind to the results when undertaking Phase 2.
7	Could you please clarify what it’s meant by the following: “The Q-CHAT was re-administered prior to any of the assessment battery.”?	We asked the parent/caregiver to complete the Q-CHAT again just before starting the assessment battery, in order to enable us to evaluate test retest reliability in a sample enriched with high scorers. This has been clarified as follows: The Q-CHAT was completed a second time prior to any of the assessment battery in order to later examine the test retest reliability of the Q-CHAT in a sample enriched with high scorers, which will be reported separately.
8	The authors employed a very conservative approach for dealing with missing data, however I am not sure how this was later incorporated in the analysis. In other words, how was movement across score groups from observed and maximum score actually taken into account (as implied in the text – end of Phase 1 Sampling strategy section)?	The full sampling strategy with study numbers is provided in Table 1 and an example has been added into the text to help with clarity (as per reviewer 2’s query). Using the minimum and maximum scores were only used for sampling, for the analysis the observed score was used, as would work in practice.
9	Do the authors believe there is scope for analysing the Q-CHAT scores in their numerical format as opposed to categories? It might worth briefly saying something about this if you think appropriate (I would certainly like to know your thoughts).	The ROC curve undertakes an analysis using each point on the Q-CHAT score individually. For screening purposes there needs to be a dichotomy of what is considered a screen positive value versus a screen negative value. Whilst we agree with the reviewer that the Q-CHAT score is a scale we are not sure how as a continuous value it can be used as a simple diagnostic tool.
10	SF1 and numbers at the start of Phase 1 Results section do not match up: 223 selected and 32 not consented vs 213 and 22 respectively in SF1.	Thank you for pointing out our error – the numbers in supplementary figure 1 were incorrect and have been corrected.
11	It’s not clear to me what the authors mean by “counterbalance administration of the ADOS-G and ADI-R.”	This means that we were unable ensure half the sample was administered the ADOS first, while the other half was administered the ADI-R first.

12	The type of graphs used in Figure 3 are not generally very commonly seen and readers might struggle to read them efficiently. Could these be turned into (or supplemented by) line diagrams across the different phases for those that were assessed at both phases? I.e. phase 1 and 2 on the x axis, different lines for typical, atypical and autism and two y axis, one for QCHAT score at phase 1 and another for phase 2?	We are not sure what the author means by a line diagram. The figures show the distributional spread of the values of Q-CHAT, CAST and the relationship between Q-CHAT and CAST. Whilst violin plots are not commonly used, they are increasingly becoming so and they do have the traditional box plots within them. We have added an additional figure (supplementary figure 4) that shows the scores based on both Phase 1 and Phase 2 diagnostic groups. We prefer to keep the original picture as some of these groups are small (or do not exist).
13	The final paragraph feels like an odd way to conclude this paper. Its content is useful and interesting, but I believe it would fit better earlier in the Discussion section to give space for the concluding remarks of the paper that relate directly to the original research question.	Thank you for this suggestion. We agree and we have re-structured the discussion, ending with a reflection about the current study and its implications.
14	Please expand the “DSM-III-R” acronym in the Introduction section	Thank you, this has been added.
15	“Luton and Bedfordshire, and Cambridgeshire” -> changed to “Luton, Bedfordshire and Cambridgeshire” or is “Luton and Bedfordshire” the preferred official term of this county/area?	Thank you for this comment. Luton is a unitary authority within Bedfordshire, so we have purposely chosen to use Luton and Bedfordshire, and Cambridgeshire.
16	I recommend “...at the specified Primary Care Trusts...” is replaced by “...at 3 Primary Care Trusts: Luton, Bedfordshire and Cambridgeshire...”	Thank you, this has been amended.
17	Replace “...and the sampling strategy was applied...” by “...and the sampling strategy as detailed below, was applied...”	Thank you, this has been amended.
18	What does random.org 8 stand for/mean?	Apologies, there was a typo here. We have clarified what this is in the manuscript as follows: selection for assessment was undertaken using a random number generator (www.random.org)
19	“...questionnaires were returned: 436 from Luton...” – notice the newly added colon	Thank you, this has been added.
20	Add a footnote on table 2 to explain the acronyms, TP, FN, FP, TN, PPV, NPV, ROC for the readers’ benefit. Similarly, for Figure 1 to explain ASC.	Thank you, this has been added. We have changed ASC to autism in Figure 1. We have additionally changed the abbreviation ROC to reflect the area under the ROC curve (AUC).
21	Finally, I personally found the use of the word administer/administration when referring to the use of	Thank you. We agree and have changed all instances to completion/completed etc.

	questionnaires/interviews, etc a bit odd but I appreciate this is very subjective.	
	Associate Editor	
1	Reviewers have provided plenty of praise for this manuscript, which reports a well conducted study. The revisions requested are more minor than major - but there are quite a few of them I have classified this as a major request. However, we would welcome an amended version of this paper.	Thank you for your kind comments and for the opportunity to revise this paper.
	Editor in Chief	
1	What this study adds please rephrase. The 3 statements below are taken from your discussion and summarise your findings better. I suggest replacing your existing text with the statements below 1. It is possible to detect autism at 18–30 months 2. It is not possible to detect every child at a very young age who will later be diagnosed as autistic. 3. The Q-CHAT can be used to screen toddlers at 18–30 months.	Thank you for this suggestion, this has been amended.

Yours sincerely,

Carrie Allison

VERSION 2 – REVIEW

REVIEWER	Reviewer name: Dr. Serena Petrocchi Institution and Country: Not applicable Competing interests: None
REVIEW RETURNED	08-Feb-2021

GENERAL COMMENTS	Dear Authors, I have read the paper for a second time and I found that every my comments or required revisions addressed. Therefore, I suggest to publish the paper in the present form. Many thanks for the opportunity.
---

REVIEWER	Reviewer name: Dr. Lonnie Zwaigenbaum Institution and Country: University of Alberta Faculty of Medicine and Dentistry, Pediatrics, Canada Competing interests: None
REVIEW RETURNED	17-Feb-2021

GENERAL COMMENTS	I appreciated the opportunity to review this revision. The authors have addressed many of the points raised in the first set of reviews. This remains an admirable study, with considerable translational value. However, the manuscript also continues to be a challenging read. This
--

	in part reflects the complexity of the undertaking, but there could also be further clarity regarding the specific study aims and how the details of the methodology map onto these aims. Specific comments and recommendations are as follows: Introduction  - Page 32 (from version with track changes), line 19: in response to Reviewer 2, Comment 2, the authors note in relation to the previous Q-CHAT study, 'test accuracy data were not collected...' Please clarify; do you mean 'predictive accuracy', or 'accuracy in a screening context'? Discrimination between autistic and non-autistic toddlers also relates to accuracy, but I think what is meant is that the Q-CHAT has not previously been evaluated as a prospective screen. - Page 33, line 34-43. Should elaborate on the aims of Phases 1 and 2. Phase 1 includes more than assessment of test accuracy (i.e., extends to selection of an optimal cut-point), and clearly Phase 2 includes more than just 'identifying autistic children not detected in Phase 1'. Whether here or in the Methods section, there should be a more exhaustive list of aims. This would help the reader understand the rationale for the analytic approach (e.g., there is only brief mention under Analytic Approach to 'a series of sensitivity analyses to decisions on the screening instrument were undertaken'; p 39, line 20-22, and Table 2 is difficult to follow as a result). Clearly delineated aims would also help with organizing the Discussion (see below). Methods  - Page 33, line 49-55 ('Phase 1 Sampling Strategy'). Even with the reviewers' thoughtful responses to the previous reviews, I still found this confusing. Further edits are needed to further differentiate between the strategy for identifying the optimal cut-point on the Q-CHAT for screening, and that for selecting participants for follow-up assessment. Further clarification is also needed as to whether calculation of observed vs. maximum scores was primarily an analytic strategy to guide stratification of participants for follow-up, vs. how individual risk might be assessed. - Page 38. Similarly, Table 1 should be titled, 'Phase 1 Follow-up Assessment Sampling Strategy' ('Follow-up Assessment 'added) Results  - Page 39, line 52: Although the rationale was discussed in the Response, 'possible autism' should be defined in the Methods section, before being introduced here - Page 42, line 41: Per earlier comment about clarifying the specific aims, what is the rationale for reporting outcome of Phase 2 participants who did not participate in Phase 1? Does not inform the stability of ASD diagnoses from Phase 1, and the authors indicate that they will report screening features of this group in another paper. - Even with the clarification in the Response, should explain why a screening cut point of 39 was selected. It may be evident from the Supplemental materials, but should provide some rationale in the main document. Discussion  - As suggested earlier, the discussion could be more explicitly aligned to the specific aims of the manuscript. The added text at the end provides a helpful summary of the clinical implications.
--	---

VERSION 2 – AUTHOR RESPONSE

Dear Dr Lucas and Prof Choonara,

Re: Manuscript ID bmjpo-2020-000700 - "The Quantitative Checklist for Autism in Toddlers (Q-CHAT). A population screening study with follow-up: the case for multiple time–point screening for autism."

Thank you very much for providing us the opportunity to make these revisions. We are grateful for the careful review that the editors and reviewers have given to our paper. The table below summarises how we have addressed the comments/reviews in the manuscript.

	Reviewer comments	Amendments
	Reviewer #2	
1	Page 32 (from version with track changes), line 19: in response to Reviewer 2, Comment 2, the authors note in relation to the previous Q-CHAT study, ‘test accuracy data were not collected...’ Please clarify; do you mean ‘predictive accuracy’, or ‘accuracy in a screening context’? Discrimination between autistic and non-autistic toddlers also relates to accuracy, but I think what is meant is that the Q-CHAT has not previously been evaluated as a prospective screen.	We have amended the text as follows to make it clearer: The Q-CHAT was not evaluated as a prospective screen in this sample.

	Reviewer comments	Amendments
2	Page 33, line 34-43. Should elaborate on the aims of Phases 1 and 2. Phase 1 includes more than assessment of test accuracy (i.e., extends to selection of an optimal cut-point), and clearly Phase 2 includes more than just ‘identifying autistic children not detected in Phase 1’. Whether here or in the Methods section, there should be a more exhaustive list of aims. This would help the reader understand the rationale for the analytic approach (e.g., there is only brief mention under Analytic Approach to ‘a series of sensitivity analyses to decisions on the screening instrument were undertaken’; p 39, line 20-22, and Table 2 is difficult to follow as a result). Clearly delineated aims would also help with organizing the Discussion (see below).	We have amended to text to: The objective of this study is to report a population screening study of nearly 4,000 toddlers at 18–30 months using the Q–CHAT with diagnostic assessments on a sub-sample of responders, and subsequent follow–up at four years. The child health surveillance database was used to identify the population eligible to screen with the Q-CHAT. The study is reported in two phases, as undertaken. The aims of Phase 1 were: i) to determine test accuracy of the Q–CHAT in the toddler period and; ii) to determine an optimal screening cut-point that could be used to select toddlers for a diagnostic assessment. The aims of Phase 2 were i) to rescreen the population at a minimum age of four using the Childhood Autism Spectrum Test (CAST)³⁷ in order to identify autistic children who were not detected at Phase 1 (false negatives) and the Checklist for Referral (CFR) to seek information on the child’s history of developmental concerns; ii) to confirm the outcome of those who were detected by the Q-CHAT at 18 – 30 months, and those who were not, by conducting diagnostic assessments; iii) to assess the stability of the

diagnostic outcome from Phase 1 to Phase 2, and; iv) to assess the discriminant power of the Q-CHAT and CAST and CFR in distinguishing autism cases from non-autism cases. This two-phase design allowed us to report sensitivity, specificity and PPV. Test accuracy at Phase 1 was not evaluated at the time to ensure researchers remained blind to the results when undertaking Phase 2.

We have also added the following text about the sensitivity analyses: At both phases, these included examining the cut-point at 31 and 38, as well as the cut-point of 39 including those parents that reported concerns, and 39 adjusted for initial non-response at screening.

	Reviewer comments	Amendments
3	Page 33, line 49-55 ('Phase 1 Sampling We have amended the text to differentiate Strategy). Even with the reviewers' more clearly between identifying the cut- thoughtful responses to the previous reviews, I still found this confusing. Further edits are needed to further differentiate between the strategy for identifying the optimal cut-point on the Q-CHAT for screening, and that for selecting participants for follow-up assessment. Further clarification is also needed as to whether calculation of observed vs. maximum scores was primarily an analytic strategy to guide stratification of participants for follow-up, vs. how individual risk might be assessed.	point and identifying children for a follow-up assessment: The strategy for selecting participants for follow-up assessment was weighted towards those with higher scores. The rationale for choosing 44 as the cut-point for inviting participants for follow-up assessment at Phase 1 was based on anticipated estimates of prevalence of autism (approximately 1%) balanced with the knowledge that over 70% of the children who already had a diagnosis on the autism spectrum scored 44 and above in our initial study³⁰. 44 was therefore chosen in order to maximise sensitivity whilst generating a feasible number of assessments to be completed. In contrast, the strategy for identifying the optimal screening cut-point on the Q-CHAT was not determined until after Phase 2 diagnostic assessments were complete, which was one the of the aims of the study. At a cut-point of 44 on the Q-CHAT at Phase 1, 100% of children were selected for assessment. We have also clarified the description of our

missing data strategy as follows: The rationale for incorporating missing data into the sampling strategy was to ensure that children with a high likelihood of having many autistic traits (and therefore potentially being autistic) but who had sufficient missing data to put them in a sampling band where the chance of being selected was low, were included in the follow up assessments.

3 Page 38. Similarly, Table 1 should be titled, 'Phase 1 Follow-up Assessment Sampling Strategy' ('Follow-up Assessment 'added)

Thank you, this has been amended to: Phase 1 Follow-up Assessment Sampling Strategy, with study numbers

	Reviewer comments	Amendments
4	Page 39, line 52: Although the rationale was discussed in the Response, 'possible autism' should be defined in the Methods section, before being introduced here	We have added clarification about our use of the word 'possible in the methods, in the section entitled Diagnostic Outcome, as follows: We used the term 'possible' to reflect the reluctance of some clinicians to commit to an autism diagnostic label at such an early age.
5	Page 42, line 41: Per earlier comment about clarifying the specific aims, what is the rationale for reporting outcome of Phase 2 participants who did not participate in Phase 1? Does not inform the stability of ASD diagnoses from Phase 1, and the authors indicate that they will report screening features of this group in another paper.	We appreciate the word 'took part' could be misleading. All the children who had diagnostic assessments at Phase 2 had completed the Q-CHAT at Phase 1. Therefore the rationale for including participants at Phase 2 who did not have a diagnostic assessment at Phase 1 was to determine the missed cases. The paragraph has been amended to make it clearer as follows: 81 children were assessed at both Phase 1 and 2. Of the 9 children whose diagnostic outcome at Phase 1 was autism who had a diagnostic assessment in Phase 2, all were still classified as autism. A further 6 children were now classified as autism (4 who were typical at Phase 1, and 2 who were atypical). 77 children were assessed for the first time at Phase 2, having completed a Q-CHAT at Phase 1. Some of these children were invited for assessment due to indicating on the CFR

		that they had received a clinical diagnosis of autism, and others because of their CAST score (15 or above).
6	Even with the clarification in the Response, should explain why a screening cut point of 39 was selected. It may be evident from the Supplemental materials, but should provide some rationale in the main document.	We have added a further clarification, as follows: The cut-point of 39 was determined by the data obtained at Phase 1 and Phase 2. We selected participants across the threshold of 44 following the Q-CHAT at Phase 1 in varying proportions in order to accurately determine where the cut-point should lie. Examination of the Q-CHAT data against diagnostic outcome, suggested that the cut-point of 39 is the point that maximises sensitivity and specificity at 18 months.

7 As suggested earlier, the discussion could be more explicitly aligned to the specific aims of the manuscript. The added text at the end provides a helpful summary of the clinical implications.	The first paragraph of the discussion has been re-organised around the aims of the study, as follows: The first two aims of the study were to determine the optimal screening cut-point on the Q-CHAT, and to determine the test accuracy. At Phase 1 outcome, all children who were classified as possible autism failed the Q-CHAT at a cut-point of 39, demonstrating that early detection and diagnosis of autism is possible. The PPV of the Q-CHAT for autism was 17%. Misinterpretation of behaviours that are not well-established as well as developmental immaturity may contribute to low PPV. An aim of Phase 2 was to rescreen the population at a minimum age of four using the Childhood Autism Spectrum Test (CAST) in order to identify autistic children who were not detected at Phase 1 (false negatives) and the Checklist for Referral (CFR) to seek information on the child's history of developmental concerns and diagnoses. The results indicated that the Q-CHAT did not identify all children who were later diagnosed as autistic by age four. We therefore were able to confirm the outcome of those who were detected by the Q-CHAT at 18 – 30 months, and those who were not. A further aim at Phase 2 was to confirm the
---	---

discriminant power of the Q-CHAT and CAST and CFR in distinguishing autism cases from non-autism cases. At a cut-point of ≥ 39 , the sensitivity of the Q-CHAT to predict autism was 44% (95%CI 26%–64%), specificity was 98% (95%CI 97%–99%), and PPV was 28% (95%CI 15%–46%). This demonstrates that the Q-CHAT alone cannot be used to detect all autistic children. This might reflect the Q-CHAT properties, but equally it likely reflects a subgroup of autistic children who do not show symptoms of sufficient 'severity' until later in childhood, and/or that symptoms of autism change with age, so require developmentally sensitive instruments at different ages. DSM-5's caveat that symptoms might not fully manifest until social demands exceed capacities may apply

	Reviewer comments	Amendments
		here. The CAST picked up many of the cases of autism that were missed by the Q-CHAT, demonstrating the need for repeat screening rather than relying on a single instrument at one time point. However, the sensitivity of the CAST at a cut point of 15 was only moderate (56%), compared to our earlier study showing sensitivity of 100%.³⁷ All children who participated in Phase 2 assessments who were identified as 'possible autism' at Phase 1 were still classified as autistic, suggested the stability of the diagnosis from Phase 1 to Phase 2.
	Associate Editor	
1	Reviewer 2 suggests that the aims and relationship between phase 1 and 2 are still not entirely clear. I found the aims of each phase on page 7/33 clear but you might consider whether adding a series of research question within these aims would be useful.	This has been done, as suggested by Reviewer 2.
2	My view is that the figure are difficult to follow, and that attention to these would be worth while because it will help your research to be understood. The version I have lacks titles and legends for figures, which may be a production issue but	We have revised Figures 1 and 3. Figure 1 simply summarises the study design. Figure 3 shows the participant study flow through screening and diagnostic assessments from Phase 1 to Phase 2, which we hope is clearer and easier to follow. Figure legends have

doesn't help with clarity. But I also found it somewhat difficult to follow the phasing of the study. The layout of the figures doesn't show the flow of participants between studies e.g. in Figure 1 it looks as if they were concurrent which I know they weren't. Figure 3 could be helpful, but only if it shows all participants as they flow sequentially through each step of the process shown in 1.

been added and revised.

	Reviewer comments	Amendments
3	It would be helpful if authors could attend to these figures to help the reader. I suggest Figure 1 shows only study phasing and the relationship between the research steps, so reader can easily understand the process. Then Figure 3 should begin with the total population sent the Q CHAT, the proportion returned, proportion consenting, should be amended to include Phase 2 and to include children stepped out of your sample (ie a side branch for those children not returning Phase 2 materials, another for not selected for further assessment at Phase 2).	This has been done, as suggested.
4	More complete labelling would also help, such that these figures have titles and legends such that they are comprehensible in a stand-alone representation.	Labelling has been added to the figures.
5	I think a proper study diagram, and some attention to language (in the methods you use the term “diagnostic assessment” but this becomes “assessment” in the results section. The former is better.	I hope Figure 3 captures this request. All instances of ‘assessment’ have been amended to ‘diagnostic assessment’ as appropriate.
6	Reviewer 2 makes some additional suggestions which authors may wish to respond to.	These suggestions have been incorporated into the revised manuscript.

	Editor in Chief	
1	I found Figure 2 hard to understand. Please expand accompanying text. Supplementary Fig 4 is also confusing	The text describing the figure has been amended to: See Figure 2 for characteristics and comparison of the Q-CHAT at Phases 1 and 2, and the CAST at Phase 2 (not weighted), according to diagnostic outcome. Supplementary figure 4 was added at the request of reviewer 3 in the last round of revision, and shows the Q-CHAT scores based on both Phase 1 and Phase 2 diagnostic groups. We are happy to leave this out if this is confusing.

Yours sincerely,

Carrie Allison
Carrie Allison
